Gut microbiota differences between paired intestinal wall and digesta samples in three small species of fish

Nyholm Lasse
Odriozola Iñaki
Martin Bideguren Garazi
Aizpurua Ostaizka
Alberdi Antton antton.alberdi@sund.ku.dk
Center for Evolutionary Hologenomics, GLOBE Institute, University of Copenhagen , Copenhagen , Denmark
Nowak Barbara
Electronic publication date: 2022 Feb 22
Publication date: 2022
Volume: 10
Electronic Location ID: e12992
Received 2021 Nov 11; Accepted 2022 Feb 2
Copyright: ©2022 Nyholm et al.
Copyright year: 2022
Copyright holder: Nyholm et al.
License: This is an open access article distributed under the terms of the Creative Commons Attribution License, which permits unrestricted use, distribution, reproduction and adaptation in any medium and for any purpose provided that it is properly attributed. For attribution, the original author(s), title, publication source (PeerJ) and either DOI or URL of the article must be cited.
License URL: https://creativecommons.org/licenses/by/4.0/

Keywords: Host-microbiota interactions, Fish microbiome, Allochthonous, Autochthonous

Funding: National Research Foundation DNRF143 Lundbeckfonden R250-2017-1351 Ostaizka Aizpurua and Antton Alberdi received the Danish National Research Foundation grant DNRF143. Antton Alberdi also received the Lundbeckfonden grant R250-2017-1351. The funders had no role in study design, data collection and analysis, decision to publish, or preparation of the manuscript.

==============================
The microbial gut communities of fish are receiving increased attention for their relevance, among others, in a growing aquaculture industry. The members of these communities are often split into resident (long-term colonisers specialised to grow in and adhere to the mucus lining of the gut) and transient (short-term colonisers originated from food items and the surrounding water) microorganisms. Separating these two communities in small fish are impeded by the small size and fragility of the gastrointestinal tract. With the aim of testing whether it is possible to recover two distinct communities in small species of fish using a simple sampling technique, we used 16S amplicon sequencing of paired intestinal wall and digesta samples from three small Cyprinodontiformes fish. We examined the diversity and compositional variation of the two recovered communities, and we used joint species distribution modelling to identify microbes that are most likely to be a part of the resident community. For all three species we found that the diversity of intestinal wall samples was significantly lower compared to digesta samples and that the community composition between sample types was significantly different. Across the three species we found seven unique families of bacteria to be significantly enriched in samples from the intestinal wall, encompassing most of the 89 ASVs enriched in intestinal wall samples. We conclude that it is possible to characterise two different microbial communities and identify potentially resident microbes through separately analysing samples from the intestinal wall and digesta from small species of fish. We encourage researchers to be aware that different sampling procedures for gut microbiome characterization will capture different parts of the microbiome and that this should be taken into consideration when reporting results from such studies on small species of fish.

Introduction

Microbes found in the gastrointestinal (GI) tract of vertebrates can play an important role in shaping their host’s fitness (Barko et al., 2018; Compant et al., 2019; Legrand et al., 2020). Hence, the study of these microbes and how they interact with their respective hosts are of high priority for a better understanding of basic ecology questions as well as its implication in many applied fields of research (Nyholm et al., 2020). However, some microbes are likely to influence their host more than others based on how long they are found in the gut and how intimately they are connected with the mucosa and mucus layer of their host (Kim, Brunt & Austin, 2007; Ringø et al., 2016; Gajardo et al., 2016). As a consequence, microorganisms found in the intestinal tract of fish are often classified as transient (allochthonous) or resident (autochthonous) (Ringø et al., 2003; Kim, Brunt & Austin, 2007). Transient microbes are considered as “short-term colonisers” passing through the gastrointestinal tract and are to a high degree influenced by environmental factors (Legrand et al., 2020). In contrast, resident microbes are considered to be more or less constantly present on the mucus surface of the intestine or in close proximity to epithelial cells (Ringø et al., 2016) and believed to be affected by host genotype (Llewellyn et al., 2014; Bolnick et al., 2014; Legrand et al., 2020) and the physical and chemical environment in and on the mucus layer (Nayak, 2010). Previous studies addressing these two communities in teleost fish have found that both communities are dominated by Proteobacteria, yet the resident community is less diverse than the transient community (Kim, Brunt & Austin, 2007; Wu et al., 2010; Gajardo et al., 2016; Riiser et al., 2018). As a consequence, transient and resident microbial communities are likely to vary, not only in their diversity, composition and abundance profiles, but also by the magnitude of which they are influenced by their host and vice versa (Kim, Brunt & Austin, 2007; Gajardo et al., 2016).

Setting a threshold for when a microbe is considered a transient visitor or a resident part of the host-associated microbiome is not straightforward, but specific sampling protocols can be employed for a better approximation to one or the other. For larger fish (e.g., Salmonids) it is common practice to collect the intestinal digesta to characterise the transient microbial community and then wash the intestine wall with saline buffer and then collect mucosal scrapes or swaps to characterise the resident microbial community (Kim, Brunt & Austin, 2007; Feng et al., 2010; Gajardo et al., 2016). This is more difficult when handling smaller fish (e.g., zebrafish) as it is rarely possible to open the intestine longitudinally and wash the intestinal lining with saline buffer due to the size and fragility of the intestinal tissue. As a consequence, researchers working with small species of fish or early life stages of larger fish will often sample the whole intestine including digesta (Burns et al., 2017; Breen et al., 2019; Almeida, Domingues & Henriques, 2021), which hampers the distinguishing between transient and resident microbial communities. One solution to reduce the impact of transient microbes when characterising the resident microbiome of small captive fish is to collect the samples after a period without access to food (Lan & Love, 2012; Sullam et al., 2012). However, this is not feasible when working on wild fish, as the chance of distorting the “wild” microbiome is likely to increase with time an animal spends in captivity (Restivo et al., 2021a) and time without access to food (Xia et al., 2014; Kohl et al., 2014; Mekuchi et al., 2018; Li et al., 2019). An alternative method is to squeeze the digesta out of the intestine and store the empty intestine (resident community) and digesta (transient community) separately.

To investigate the ability of the latter described sampling procedure to identify the transient and resident microbial communities of small fish species or early life stages of larger fish, we collected paired samples of the intestinal wall (a proxy for mucus-mucosa-resident community) and digesta from three small cyprinodontiform fish species from different locations. We used 16S amplicon sequencing to examine if this sampling protocol is able to identify whether (1) samples from the intestinal wall and digesta differ in their diveristy of microbes (2) if the community structure is different between the intestinal wall and digesta and (3) if it is possible, even with spillover between sample types, to identify taxa that are likely to be a part of the resident microbiota.

Materials and Methods

Sample collection and storage

Spanish toothcarps (Aphanius iberus, AI; n = 27), Eastern mosquitofish (Gambusia holbrooki, GH; n = 26) and Valencian toothcarps (Valencia hispanica, VH; n = 23) were caught using baited minnow traps at 11 wetland locations in Valencia, Southeastern Spain, in August 2017 and September 2018 (Table S1) with permission from the Government of Valencia. These three species of fish were chosen as part of a larger project investigating the influence of the microbiome in species conservation, and because they serve as good representatives of the challenges faced by researchers when sampling the gut microbiome of small wild fish species. To minimise contamination risk during dissection, fish were taken to a research facility while keeping them in large (40L) oxygenated tanks separated by species and location with water from their natural habitat for a maximum of four hours.

Right before dissection, fish were euthanized by a quick blow to the head in accordance with the Spanish law on animal research ethics (RD 53/2013) and the European Directive on the protection of animals used for scientific purposes (2010/63/EU). The entire intestine of each fish was extracted using scalpels and forceps in a sterilised environment and two types of samples were obtained (Fig. 1). Digesta samples (D, Fig. 1) were collected by squeezing the intestine using forceps and stored in absolute ethanol in individual tubes. As very little digesta was present in the samples, we used all available digesta from each fish as input for the extractions. The entire emptied intestines were used as a proxy for the intestinal wall microbial community (W, Fig. 1) and were cut into smaller pieces using sterile scalpels over sterile weighing boats and stored in ethanol. Animals from which it was not possible to squeeze out any digesta of the dissected intestine were excluded from this study and are not a part of the total reported sample size. Samples were stored at −20 °C until DNA extraction.

Figure 1 Overview of sample types.

The two sample types used in the study are denoted either by a capital W (intestinal wall) or D (digesta) and an overview of the number of species, locations, samples, length and weight.

DNA extraction

DNA was extracted using the tube-based Powersoil® DNA Isolation Kit (MoBio, CA, USA) following the manufacturer’s protocol with minor modifications (Table S2). DNA was eluted in 50 µl EB buffer and stored at −20 °C. Extraction blanks were included in each extraction round to screen for potential contamination. All extractions were done in a dedicated pre-PCR laboratory.

PCR, library build and sequencing

The generation of sequencing data and its subsequent bioinformatic processing were done similar to what has previously been described in Aizpurua et al. (2021). Specifically, PCR amplification of the V3–V4 region of the bacterial 16S rRNA gene was done using a broadly validated primer set (341F-805R) (Muyzer, De Waal & Uitierlinden, 1993; Caporaso et al., 2011). Addition of 24 different tags to the 5′-end of both primers enabled the differentiation of pooled samples after sequencing (Binladen et al., 2007). For each sample three technical replicates were PCR amplified to minimise the effect of PCR stochasticity (Alberdi et al., 2017). PCR amplifications were done using an Applied Biosystems 2720 Thermal Cycler with a total reaction volume of 25 µl consisting of 13.5 µl ddH2O, 2.5 µl (10×) Gold Buffer (GeneAmp®), 2.5 µl (25 mM) MgCl2, 1.5 µl (20 ng/µl) BSA, 0.5 dNTPs (10 mM), 0.5 µl (5 U/µl) DNA polymerase (AmpliTaq Gold®), 2 µl (10 mM) primer mix (forward and reverse) and 2 µl DNA extract. PCR settings were 95 °C for 10 min, 30 cycles of 95 °C for 15 s, 53 °C for 20 s and 72 °C for 40 s and 72 °C for 10 min. PCR products were visualised on 2% agarose gel by loading a mix of 4 µl PCR product with 2 µl loading buffer and given a score (0–3) based on band strength. Based on the band strength scores (0 = 10 µl, 1 = 10 µl, 2 = 8 µl and 3 = 10 µl) the PCR products were pooled together in pools consisting of 24 unique tags allowing the tracking of each sample back to individual fish (Binladen et al., 2007). Amplicon pools were subsequently purified using SPRI beads (DeAngelis, Wang & Hawkins, 1995; Rohland & Reich, 2012) in a 1:1 beads:DNA ratio to remove non-target DNA and primer dimers. Amplicon pools were quantified on a Qubit (Thermo Fisher Scientific®) and diluted in 30 µl water to a total amount of 500 ng of DNA. Libraries were constructed using the Blunt-End Single-Tube Tagsteady protocol described by Carøe & Bohmann (2020) and quantified through qPCR by mixing 2 µl 1:10,000 dilution of each library with 8 µl Quant Mastermix with added primers (NEB) and running it for 1 cycle of 95 °C for 1 min, 35 cycles of 95 °C for 15 s and 63 °C for 45 s on a Stratagene Mx3006P (Agilent Technologies©). Different volumes of each library were pooled together to ensure equal molarity and sequenced on an Illumina MiSeq platform aiming at 9,000–62,000 reads per PCR replicate.

Bioinformatics

Paired-end reads were demultiplexed based on library indices and trimmed using AdapterRemoval 2.2.2 (Schubert, Lindgreen & Orlando, 2016) by removing adaptor sequences, sorting reads based on tag combinations and filtering out stretches of low quality bases (Q < 30). Additionally, reads containing >5 low quality bases after trimming and reads with tags having >2 mismatches to barcodes were removed. A complete overview of extraction- and sequencing batches, sample IDs and read names can be found in Table S3. Technical replicates for each sample were merged and primer sequences removed, allowing a maximum error rate of 15% using Cutadapt 2.10 (Martin, 2011). To ensure the same directionality of sequences (note that an adapter-ligation based PCR-free library preparation was used), reads sequenced in the reverse-forward direction were reversed based on primer location. All downstream filtering, analyses and original illustrations were performed in Rstudio/v.1.2.5033 (RStudio Team, 2019). Reads were filtered based on quality and specific error signatures error models of each respective round of sequencing using DADA2 (Callahan et al., 2016). ASVs were inferred, forward and reverse reads merged with a minimum overlap of 5 bp and the taxonomy was assigned with the Silva/v132 taxonomic database (Quast et al., 2013) using DADA2. Finally the five datasets from each respective sequencing rounds were merged for further post-clustering filtering and chimeric sequences removal using DADA2. LULU (Frøslev et al., 2017) was used for post-clustering removal of erroneous ASVs. To identify samples not reaching a diversity plateau due to sequencing depth, rarefaction curves were plotted. Decontamination of potential contaminant ASVs were done using Decontam (Davis et al., 2018) with the prevalence method and default settings. This resulted in the removal of 19 contaminant ASVs with an average relative abundance across species and sample types of 0.0001 ± 0.0005 (Table S4). All ASVs identified as belonging to Vertebrates at phylum level or mitochondria at family level were removed, as well as all ASVs mapping to Chloroplast at order level as these might originate from ingested plant material (Hanshew et al., 2013). Next, the dataset was split into sub-datasets corresponding to each of the three species. To remove the impact of low-abundance and low-prevalence microbes, only ASVs present in ≥2 samples and with a relative abundance >0.1% in at least one sample were used in downstream analyses. Phylogenetic trees used in the analysis of diversity and compositional variation were generated by aligning ASVs using Clustal-Omega (Sievers et al., 2011) and then RAxML-NG (Kozlov et al., 2019) were used to infer a maximum likelihood tree.

Statistical analysis

The following analyses were done using the R packages phyloseq (McMurdie & Holmes, 2013), ggpubr (Kassambara, 2020), hilldiv (Alberdi & Gilbert, 2019), nlme (Pinheiro et al., 2021), stats (R Core Team, 2020), vegan (Oksanen et al., 2018) and Hmsc (Tikhonov et al., 2020). Functions used in their respective packages will be referred to as “package::function”.

Analysis of diversity

Paired diversity plots for each of the three species were generated using ggpubr::ggpaired. Diversity analyses were done using the Hill numbers framework (Chao, Chiu & Jost, 2014). The diversity profiles for all locations across all three species were plotted using hilldiv::div_profile (Fig. S1). The steep decline in the effective number of ASVs going from a q-value of 0 to 1 indicates a high level of unevenness between ASVs. Hence, a q-value of 1 was chosen for Hill-numbers based downstream diversity analyses, so that ASVs were weighed according to their relative read abundances (Jost, 2006). Furthermore, to account for the phylogenetic relatedness between ASVs, we also included the generated phylogenetic trees in the analysis. To satisfy the assumptions of the linear mixed-effects model, two paired samples were visually identified as outliers by plotting the residuals and subsequently removed from the AI data set (Fig. S2). Diversity of individual samples was computed using hill_div::div_hill. A non-parametric Kruskal-Wallis test was used to test for differences in diversity across fish species for each of the two sample types followed by a post-hoc Dunn test with Benjamini–Hochberg correction as implemented in hill_div::div_test. Linear mixed-effect models (nlme::lme) were used to examine how sample type influences the diversity of individual fish gut microbiomes using the Restricted Maximum Likelihood (REML) method for parameter estimation. Due to the natural interdependency of paired intestinal wall-digesta samples and individual fish within locations, Fish_ID was nested inside Locations and both were treated as random effects and Sample.type as a fixed effect. Assumptions were verified by plotting the residuals.

Compositional variation

Compositional differences were measured by means of Jaccard-type overlap metric, including phylogenetic relationships between ASVs (Chiu, Jost & Chao, 2014; Chao et al., 2019; Alberdi & Gilbert, 2019) and computed using the function hill_div::pair_dis. Based on the generated dissimilarity matrix, phyloseq::ordinate was used to create non-metric multidimensional scaling (NMDS) ordinations for the three species and phyloseq::plot_ordination was used to generate plots. To test the null hypothesis of no effect of sample type and location on bacterial community composition, a PERMANOVA analysis was done using vegan::adonis (Permanova 1) with the above-described Jaccard-type dissimilarity matrix as response. Fish_ID was used as strata to account for the dependency between intestinal wall-digesta samples within the same individual and Sample.type and Location were used as fixed explanatory factors (including their interaction). A second PERMANOVA (Permanova 2) was done to test whether the microbial community of digesta samples was influenced more by environmental factors compared to the community of the intestinal wall samples using Location as the fixed explanatory factor and either the Jaccard-type dissimilarity table for intestinal wall or digesta samples as response. For both PERMANOVAs, 999 permutations were used. The assumption of homogeneous dispersion in compared groups was tested using vegan::betadisper and stats::anova. The assumption of homogeneity of within-group dispersions was violated for locations for VH because of the presence of two individuals with very distinct microbiomes. To ensure the robustness of our results, we repeated the analysis after removing those fish with outlying values, which led to homogeneous dispersions and similar results.

To identify microbial ASVs and families enriched in intestinal wall samples as compared to digesta samples we analysed the data using a hierarchical modelling of species communities (HMSC) framework (Tikhonov et al., 2020) as implemented in the R package Hmsc (Tikhonov et al., 2021). HMSC is a class of joint species distribution models (Warton et al., 2015) to analyse multivariate community data. For that, a hierarchical model is constructed in the generalised linear model (GLM) framework and using Bayesian inference. We fitted models with two alternative response data: (1) bacterial ASV sequence counts collected in each sample type in fish individuals of the three species and, (2) bacterial families sequence counts, generated by adding up the sequences of the ASVs belonging to each family. Additionally, the three fish species were analysed in three independent models, thus making a total of six models. As the data were zero inflated, we applied a hurdle model, similar to previous publications using amplicon sequencing data (Odriozola et al., 2021). This type of model consists of two parts, one modelling the presence-absence of species and the other modelling abundance conditional on presence. To fit the first model we transformed all non-zero values in the dataset to one, to create a presence-absence matrix. Then, for the second model, we generated another dataset where we shifted all zeros to missing values, and kept all nonzeros in their values. In the presence-absence part of the model, we applied a binomial model with probit link function to each ASV, whereas to model abundances conditional on presences (scaled to mean zero and unit variance) we used the log-normal model. Then, the two components of the model were fitted consecutively. See Ovaskainen & Abrego (2020) for more details on how to apply hurdle models in HMSC to model zero inflated data.

As fixed explanatory variables in the matrix X of HMSC we included the categorical variable Sample.type, as well as the continuous variable log-transformed Sequencing.depth, which controlled for the variation in sequencing depth among samples. To account for the nested study design, we included Fish_ID and Location level random effects in the models. A significant association with a specific sample type in the binomial model means that the ASV has a higher probability of occurrence in that sample type. A significant association in the log-normal model means that, when present, the ASV is more abundant in a sample type. A posterior probability of >0.7 (<0.3 for negative association) was considered as moderate statistical support for the association with a sample type, whereas a posterior probability of >0.9 (<0.1 for negative association) was considered as strong support.

We fitted the models with Hmsc::Hmsc assuming the default priors and sampled the posterior distribution with Hmsc::sampleMcmc that ran four Marcov Chain Monte Carlo (MCMC) chains, each of which was run for 37,500 iterations, of which 12,500 were discarded as burn-in. We thinned by 100 to obtain a total of 250 posterior samples per chain and 1,000 posterior samples in total. We assessed MCMC convergence by measuring potential scale reduction factor (Tikhonov et al., 2020) for the beta parameters (measuring the response to sample type) (Fig. S3).

Results

Filtering, trimming and the removal of contaminant sequences resulted in a total of 18,882,099 merged reads. The removal of merged reads mapping to either Vertebrata (37,803 merged reads) at phylum level, Chloroplast (849,196 merged reads) at order level and mitochondria at family level (175,284) resulted in 17,819,816 remaining merged reads with all samples reaching diversity saturation in sequencing depth (Fig. S4). The removal of outliers and low abundant ASVs resulted in 474 ASVs assigned to AI, 570 ASV to GH and 114 ASVs to VH. A complete overview of post-filtering taxonomy and abundances can be found in the Supplemental Material (Data S1).

Fish species exhibited different overall microbial diversities and communities (Kruskal-Wallis test, Figs. 2A–2B, Table 1). For both intestinal wall and digesta samples, post-hoc analysis revealed that the diversity was significantly different between VH and the two other species, whereas no significant differences were found between AI and GH (Fig. 2A). Regardless of the host species, linear mixed effects models revealed that intestinal wall samples had a significantly lower diversity compared to their corresponding digesta sample for AI, GH and VH with an average diversity decline of 16%, 23% and 27% relative to the mean intercept for digesta samples, respectively (Fig. 2A).

Figure 2 Overview of effective number of ASVs and NMDS plots of dissimilarities.

(A) Boxplots showing the diversity of paired samples for each of the three species across all locations. Green lines indicate a decrease in diversity going from digesta samples to intestinal wall samples and red lines indicate an increase in diversity. Results from Kruskal-Wallis test for overall differences in diversity across species and the post-hoc Dunn tests are shown above the boxplots and results from the linear mixed-effects models (lme) are shown within each boxplot. The mean intercept refers to that of digesta samples, and the numbers over the black dotted lines are the mean slope values going from from digesta samples to their corresponding intestinal wall samples. (B) NMDS plots of dissimilarity between samples within species and (C) dissimilarity between samples across all species. Sample pairs are connected by lines, and the two types of samples are shown as either circles (digesta) or triangles (intestinal wall) and different locations are indicated by colour.

Microbial composition was significantly different between intestinal wall and digesta samples for all three species (Permanova 1, Table 1). Sample type (intestinal wall vs. digesta) was able to explain 1–12% of the variance, however the majority of the explained variance (29–49%) was attributed to differences between locations (Permanova 1, Table 1). NMDS plots supported these findings with samples from the same location clustering closer together and within locations paired samples displaying smaller, but considerable distance between sample types (Fig. 2B). The NMDS plot of the two sample types for all three species indicated that there were no general patterns in which samples of the same type clustered closer together across species and that the extent of differences between the two sample types were relative to the individual fish (Fig. 2C). We further investigated whether the relative effect of location was greater on digesta samples relative to intestinal wall samples and found location to have a significant effect on digesta samples for all three species, explaining 27–68% of the variance (Permanova 2, Table 1). Location also had a significant effect on the composition of intestinal wall samples from AI and GH, but the amount of explained variance was lower compared to digesta samples (17–48%, all species; Permanova 2, Table 1).

Table 1 Results from PERMANOVA.

	Aphanius iberus	Gambusia holbrooki	Valencia hispanica	
	R2	p-value	R2	p-value	R2	p-value	
Permanova 1							
Sample type	0.081	<0.001***	0.017	<0.001***	0.120	<0.001***	
Location	0.497	<0.001***	0.486	0.005**	0.292	<0.001***	
Sample type: Location	0.013	0.345	0.018	0.055	0.012	0.338	
Permanova 2							
Location (intestinal wall)	0.485	<0.001***	0.454	<0.001***	0.151	0.097	
Location (digesta)	0.649	<0.001 ***	0.573	<0.001***	0.252	<0.001***	
Notes.

The R2 value indicates how much of the variance is explained by a given covariate and level of significance with ** (p < 0.01) and *** (p < 0.001). Sample type: Location is the interaction between the two fixed explanatory factors.

The majority of the families were negatively associated with the intestinal wall in both binomial (AI = 90.7%, GH = 86.4% and VH = 100%) and log-normal (AI = 81.4%, GH = 92.8% and VH = 94.6%) parts of the HMSC models in AI, GH and VH: this means that most families had higher probability of occurrence and, when present, they were more abundant in digesta samples (Fig. 3, Fig. S5, Data S2). The same pattern was observed at ASV level (Data S2). Across the three species, the log-normal part of the models identified six families (AI = 4, GH = 0 and VH = 2) that, when they occurred, were more abundant in intestinal wall samples with strong statistical support (posterior probability > 0.9) and three families (AI = 1, GH = 2 and VH = 0) with moderate statistical support (posterior probability > 0.7). Some of these families were enriched in more than one fish species and in total they represented seven unique families. No families or ASVs were positively associated with the intestinal wall in the binomial part of the model (Fig. S5, Data S2). At ASV level, the log-normal part of the models found 25 ASVs (AI = 8, GH = 10 and VH = 7) that were more abundant in intestinal wall samples with strong statistical support and 58 ASVs (AI = 21, GH = 34 and VH = 3) with moderate statistical support.

Figure 3 Enriched bacterial families in the log-normal submodel.

For simplicity only families that are more abundant in intestinal wall samples compared to digesta samples with strong (red) or moderate (orange) statistical support or families more abundant in digesta samples compared to intestinal wall samples with strong (blue) statistical support are shown. Families that are significantly more abundant in either intestinal wall or digesta samples in one species of fish, but not significant in another species of fish are shown as a blank space. Plot showing all families can be found in Fig. S5.

Discussion

Amplicon sequencing of paired intestinal wall and digesta samples from the GI tract of three small cyprinodontiform fish species were able to identify two different microbial communities. We found that intestinal wall samples from the three species were significantly less diverse than digesta samples and that the overall community composition is significantly different between the two kinds of samples for all three fish species. Furthermore, we identified seven families of bacteria to be significantly more abundant in intestinal wall samples across the three species.

Overall, AI and GH had gut microbial communities with similar levels of diversity, whereas VH exhibited lower diversity (Fig. 2A). Irrespective of the overall diversity, the intestinal wall microbiota was significantly less diverse compared to the paired digesta microbiota, which is in concordance with previous studies comparing these two communities (Kim, Brunt & Austin, 2007; Gajardo et al., 2016; Nielsen et al., 2017; Riiser et al., 2018). The general higher diversity of digesta samples is expected to be a result of the mixture of environmental microbes and food items (Ringø et al., 2016; Legrand et al., 2020). We expected the intestinal wall community to also include some of these environmental microbes, as we did not wash the intestinal wall or leave the fish without access to food prior to sampling. Still, we observed a lower diversity in the intestinal wall samples, which supports the hypothesis that it takes specialised traits for a microbe to adhere and sustain a viable population in this environment (Nayak, 2010).

The analysis of between sample type dissimilarity (Permanova 1, Table 1) supports that not only are the intestinal wall community less diverse, but the community composition for all three species is also significantly different to that of the digesta. Location explained most of the variance, with substantial variation between species (Permanova 1, Table 1). Hence, we also separately examined the effect of location on the two sample types and found that location explained a larger part of the variation for digesta samples for all the included species compared to intestinal wall samples (Permanova 2, Table 1). This is in line with what has previously been proposed about the transient community being mainly influenced by environmental factors (Legrand et al., 2020) such as temperature (Martin-Antonio et al., 2007; Kokou et al., 2018), salinity (Schmidt et al., 2015), level of eutrophication (Restivo et al., 2021b), diets (Ringø et al., 2016) and the microbial communities of the water column and sediments (Minich et al., 2020). Therefore, we warrant further research into the influence of different environmental factors on both transient and resident communities as well as the influence of intrinsic factors such as genotype (Kokou et al., 2018). Furthermore, the large variation observed in the influence of location between the three species indicates that microbial communities respond differently to the influence of environmental and intrinsic factors between species of fish, which should also be the topic for further investigation.

Although the variation explained by the sample type was considerably lower than that explained by location, we detected consistent patterns of microbial enrichment between the intestinal wall and digesta across individuals and localities. A total of seven families were found to be more abundant in the intestinal wall samples compared to the digesta samples (Fig. 3). Some of these families have previously been found to include members closely associated with their host. For instance, due to their small genome sizes, members of the family Mycoplasmataceae are suspected to be intimately associated with their host and are often found to be dominating the gut microbiome of Salmonids such as rainbow trout (Oncorhynchus mykiss), Atlantic salmon (Salmo salar) and European whitefish (Coregonus lavaretus) (Brown, Wiens & Salinas, 2019; Rasmussen et al., 2021), although they also have been found to be dominating the GI tract in other fish species (Bano et al., 2007; Burtseva et al., 2021). We found Mycoplasmataceae to be enriched in intestinal wall samples from GH and VH, whereas in AI Mycoplasmataceae was enriched in the digesta sample. Interestingly, we found that ASV_1 and ASV_6, both belonging to the genus Mycoplasma, were highly enriched in intestinal wall samples of VH and GH, respectively (Data S2), but also that the same two ASVs were enriched in digesta samples from AI. This could suggest that members of this family are present in the surrounding environment, but not all of them are able to colonise the intestinal wall of all fish species. To further investigate this, future studies looking into microbial differences between intestinal wall and digesta samples should ideally include water samples collected from the different locations. A similar situation was observed for the families Brevinemataceae and Desulfovibrionaceae. Brevinemataceae have been identified in the GI tract of several fish species (Brown, Wiens & Salinas, 2019; Uren Webster et al., 2020; Iwatsuki et al., 2021) and Desulfovibrionaceae in particular have earlier been found to be enriched in samples from the intestinal wall compared to paired content samples from rabbitfish (Siganus fuscescens) (Nielsen et al., 2017). The family Microbacteriaceae has previously been identified in relation to fish (Boutin et al., 2012; Larios-Soriano et al., 2021), but as members of this family have been found on the skin and in gut digesta it seems most likely that these are transient microbes. As with Mycoplasmataceae, the small genome size of members in the family Saccharimonadaceae, enriched in AI, have led researchers to propose a symbiotic lifestyle (Lemos et al., 2019) and as a consequence we can not rule out a symbiotic relationship between members of Saccharimonadaceae and AI. The two remaining enriched families, A4b and Methylococcaceae, have to our knowledge not been identified in association with a fish host, but rather to plant root microbiomes (Vik et al., 2013; Barelli et al., 2020) and from environmental samples (Taubert et al., 2019), respectively. Even though the diversity and composition of the microbial communities of intestinal wall and digesta samples for GH were significantly different, we were unable to identify families with strong statistical support to have a higher abundance in intestinal wall samples. This could be attributed to the fragility of the GI tract of GH compared to the two other species, making it hard to squeeze out digesta without fragmenting the intestine, which could have resulted in a higher proportion of digesta left in the intestinal wall samples.

Identifying whether the members of the enriched families are truly a part of the resident microbial community in these fish will need further investigations using more detailed methods. These methods includes shotgun sequencing, which would allow researchers to look for genes within each respective metagenome assembled genome (MAG) related to host adaptations, but this method can be hampered by the large proportion of host DNA found in intestinal wall samples (Quince et al., 2017; Marotz et al., 2018). A shotgun sequencing approach can also be used to infer the level of replication of MAGs within a metagenome, which could be used as a proxy for resident time in GI tract, assuming that resident microbes will have a higher replication rate at a any given time point compared to transient microbes (Brown et al., 2016). Other alternative methods such as fluorescence in situ hybridisation (Shi, Grodner & De Vlaminck, 2021), electron microscopy (Ringø et al., 2003) and laser capture microdissection (Riva et al., 2019) can provide researchers with the spatial resolution needed to establish the level of intimacy between microbial cells and host cells.

Even though we in this study were able to find members of the microbiome that are enriched in intestinal wall samples, we are conscious that the simple sampling procedure applied in the current study is prone to yield a substantial level of spillover between paired digesta and intestinal wall samples. As a consequence we do not expect to find many families/ASVs that have a higher chance of being found in the intestinal wall compared to digesta (presence/absence) as we expect most of the spillover to originate from this sample type. Therefore, we mainly relied on the abundance data as we expected the signal to be stronger than for the presence-absence data. We observed a number of enriched families/ASVs in the intestinal wall samples that we find unlikely to be part of the resident microbial community based on previous studies on these taxa. Hence, the results should not be trusted blindly. We are aware that many of the microbes that are present in the intestinal wall samples, but not identified to be significantly more frequently found or abundant in intestinal wall samples by the HMSC models, could in fact be a part of the resident microbial community. With that in mind, it is most probable that the enrichment analysis would be able to identify more microbes to be a part of the resident microbiota if further rinsing of the empty intestine were done, but for small fish this is not always possible.

Given the nature of gut microbiome samples collected from small fish it is difficult to obtain samples that only capture microbes from either the resident or transient microbial communities. Both communities can be of relevance to answer questions regarding the overall role of the microbial gut community, but in some cases the goal is to determine which microbes can be considered as being adapted to and potentially essential for the fish host. This could for instance be the case in captive breeding of small or early life stages of endangered fish species (Lavoie et al., 2018; West et al., 2019; Abdul Razak & Scribner, 2020) or to increase the survival rate of early life stages of commercially important aquaculture species (Abdul Razak & Scribner, 2020; Borges et al., 2021). We acknowledge that due to resource limitation, in many cases it will not be possible to apply more detailed methods to identify resident microbes. By using a combination of relatively cost-efficient amplicon sequencing and conservative statistical modelling we believe it is possible to make meaningful inference regarding the origin of a given taxa, as long as results are compared and evaluated based on the findings of other studies. When it is not possible to model enrichment between paired samples, for instance when only one kind of sample is collected, we encourage researchers to be conscious of whether a given sampling procedure is likely to capture the resident or the transient community, or a mixture of the two, when reporting the results from these studies.

Conclusion

In this study we found that paired samples from the intestinal wall and digesta coming from three small fish species harbor different microbial communities. The diversity was significantly lower in samples from the intestinal wall for the three fish species and the community composition was significantly different between sample types for all three species. Using hierarchical modelling of species communities we identified a total of seven unique families to be significantly enriched in intestinal wall samples. Based on these findings we conclude that the sampling procedure used in the current study combined with conservative statistical modelling is capable of identifying host-associated microbes, but these results should be critically evaluated and that more comprehensive methods should be used to further examine the true level of host-association. When collection of paired samples is not possible, we encourage researchers to be aware that different sampling protocols are likely to capture different parts of the fish gut microbiome and that this should be incorporated in the interpretation of the results.

Supplemental Information

Table S1 Overview of sampling details for the 11 locations in Spain and length and weight of individual fish

Click here for additional data file.

Table S2 PowerSoil Extraction Protocol

Click here for additional data file.

Table S3 Overview of sequencing details for demultiplexed reads

Click here for additional data file.

Table S4 Overview of ASVs identified as contaminants by the decontam software (Callahan et al., 2018)* using default settings

Click here for additional data file.

Figure S1 Diversity profiles for the three fish species

(a) Aphanius iberus, (b) Gambusia holbrooki and (c) Valencia hispanica going from q = 0 to q = 3. The higher the q-value the more weight is added to abundant ASVs. For all three fish species we observe a large drop in the effective number of ASVs going from q = 0 to q = 1 indicating that a few ASVs are dominating the communities.

Click here for additional data file.

Figure S2 Identification of outliers in Aphanius iberus and change in residuals

(a) Plots of standardized residuals vs. fitted values including outliers and (b) after moving outliers. Removing the outliers results in a more uniform distribution around the horizontal 0 line.

Click here for additional data file.

Figure S3 Mixing statistics evaluating the convergence of the HMSC models

Convergence at (a) Family level and (b) ASV level. The plots show potential scale reduction factors for the beta parameters of the binomial (left column) and lognormal (right column) components of the models for Aphanius iberus (top), Gambusia holbrooki (middle) and Valencia hispanica (bottom). Beta parameters measure the families/ASV’s responses to explanatory variables. Potential scale reduction factors are close to 1, thus we conclude the MCMC convergence was satisfactory.

Click here for additional data file.

Figure S4 Rarefaction curves at different levels of resolution

(a) Aphanius iberus, (b) Gambusia holbrooki and (c) Valencia hispanica visualizing the saturation in number of detected ASVs versus sequencing depth.

Click here for additional data file.

Figure S5 Heatmap of the estimated parameters β (i.e. response of the families to the sample type intestinal wall with respect to digesta for different model components)

(a) Aphanius iberus, (b) Gambusia holbrooki and (c) Valencia hispanica. Red and orange boxes indicate positive significant associations, dark and light blue boxes indicate negative significant associations, and, white boxes indicate non-significant associations.

Click here for additional data file.

Data S1 Overview of taxonomy and read counts for each ASV after filtering (Valencia hispanica)

Click here for additional data file.

Data S2 Parameter estimates from intestinal wall samples at family and ASV level from the HMSC model relative to digesta samples

occ_mean is the mean parameter estimate for the occurence (binomial) model and abu_mean is the mean parameter estimate for the abundance (log-normal) based model. Support values indicate the level of statistical support, where >0.7 and <0.3 are cosidered as moderate statistical support, and >0.9 and <0.1 are considered as high statistical support for positive and negative association, respectively.

Click here for additional data file.

The authors would like to thank the staff at Centro de Conservación de Especies Dulceacuícolas de la Comunitat Valenciana, and especially Pilar Risueño and Jesús Hernández, for their indispensable help in collecting samples from the three fish species as well as providing knowledge on the biology of the species.

Additional Information and Declarations

Competing Interests

Author Contributions

Animal Ethics

Data Availability

The authors declare there are no competing interests.

Lasse Nyholm conceived and designed the experiments, performed the experiments, analyzed the data, prepared figures and/or tables, authored or reviewed drafts of the paper, and approved the final draft.

Iñaki Odriozola analyzed the data, prepared figures and/or tables, authored or reviewed drafts of the paper, and approved the final draft.

Garazi Martin Bideguren performed the experiments, authored or reviewed drafts of the paper, and approved the final draft.

Ostaizka Aizpurua and Antton Alberdi conceived and designed the experiments, performed the experiments, analyzed the data, authored or reviewed drafts of the paper, and approved the final draft.

The following information was supplied relating to ethical approvals (i.e., approving body and any reference numbers):

Government of the Valencian Community.

The following information was supplied regarding data availability:

The demultiplexed reads are available at the ENA repository: PRJEB48573.

The code is available at GitHub: https://github.com/LasseNyholm/Gut-microbiota-differences-between-paired-mucus-and-digesta-samples-in-three-small-species-of-fish.

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
