# Peer review of "Gut microbiota differences between paired intestinal wall and digesta samples in three small species of fish"

_PeerJ, doi:10.7717/peerj.12992_

## Round 0.1 · original submission · Major Revisions

This is an interesting manuscript but as you can see from the comments from both reviewers it requires a number of corrections. Please provide justification why those fish species were selected and make sure to provide all missing details, for example how the outliers were determined. The way the samples were collected on could argue that the mucus samples were really mucosa samples, as some mucus could have been removed with digesta. Please justify why you think this does not happen.

·

Basic reporting

The manuscript is well written, with an interesting topic and a logical structure. It is focused on demonstrating the baseline of the gut microbial diversity and composition variation between mucosa and digesta samples in three small species of fish. This will provide insights into fish residential gut microbiota identification. The introduction is appropriate to support the context, and the results are well discussed. If more information is provided as described below, I recommend that the manuscript can be published in Peer J.

Line 31. Please consider removing the word ‘time’ to keep consistent.
Line 64-66. Please support your statement with relevant references.

Experimental design

The research question is well defined, relevant, and meaningful. The experiment is well designed, but more detailed information in methods is needed. Genomic data analysis seems solid, but in the reviewer's opinion, the taxonomic analysis can go deeper to a further level, such as the genus level.

Line 78-81. More detailed information in terms of fish size is needed, such as fish length, weight, etc.
Line 92. If emptied intestines are not washed with saline buffer, how do you make sure there is no digesta left?
Line 152. ‘>=’ is not the correct format. Please change the symbol to ‘≥’.
Line 153-155. It stated that phylogenetic trees were generated. Please indicate the result and include it in the supplementary section at least. If this is irrelevant, you should consider removing this sentence from the manuscript.
Line 185-200. The indication of nMDS and PERMANOVA analyses belongs to the β diversity analysis. These sentences are more suitable for the last section ‘Analysis of diversity.
Line 199-200. How was the resemblance matrix of the dataset created? For example, the resemblance could be created by calculating the Euclidean distance or Bray-Curtis distance.
Line 202. The fish gut microbiome has been studied in many studies via 16S rRNA sequencing in recent years. Taxonomic composition at the family level may not be powerful or informative enough to explain the variation in the intestinal microbial community. One wonder why you only analysed the microbiota at the family level.

Validity of the findings

All underlying data have been provided. The last section in the discussion can provide insights to further studies, but a clear conclusion can be helpful for summarising the highlights.

Line 256-257. An additional nMDS figure showing the overall microbial similarity between the two sample types is recommended here.
Line 257-259. How do you identify the variance regarding sample type and location? And where is the detailed information?
Line 264. Where is the ‘Table 3’? I assume it is the attached Table 1.
Line 261 -266. This part is a bit confusing. In the discussion (Line 302), you stated the location explained most of the variance. However, the result indicated that the explained variance was lower compared to digesta samples.
Line 267-269. “Most of the families” is not precise. Please demonstrate the result in number or percentage.
Discussion
Line 285-286. The word ‘unique’ is not appropriate to be used here. From my understanding, unique families indicate the families that are either detected in mucus or digesta only. so, there is no comparative degree.
Line 287 and line 366. Subtitles are unnecessary in the discussion section.
Line 319. Remove the word ‘unique'.
Line 323. Please clarify the ‘Salmonids’ species. Mycoplasmataceae are not found to be dominating the gut of freshwater farmed Chinook salmon.
Line 330-331. Please reword this sentence. Detecting these families in digesta and mucus samples does not mean it can be detected in the surrounding environment. It can only be confirmed by analysing the microbiome from the rearing water.

Additional comments

All comments have been listed above.

Reviewer 2 ·

Basic reporting

A figure showing where the fish were sampled from along with pictures of the various species would be useful.

Figure 2 could be improved by
a) showing an NMDS with all samples
b) improving the legend to describe the various metadata categories

Experimental design

Overall the design is good. There is enough replication across species and locations to test the hypothesis if the digesta differs from the general GI tract resident community. The computation methods are generally ok as well. A few questions for clarification:
Why were these species of fish chosen? Did you measure the GI length and total length? Do they occupy different trophic niches?


Line 88 states fish with empty stomachs were excluded. How did one determine this – please explain in more detail along with the approximate proportion of fish which were excluded in the end? Also what were the sizes of the actual fish?

Its likely that during transport many of the fish would have been stressed and therefore defecated in the bags. Were fish transported in individual bags or mixed bags? If mixed, was that also tested to determine if fish from the same transport bag had a more similar microbiome? In the future, I would recommend the researchers kill the fish onsite and preserve either direct in 95% etoh at RT until processing or directly place on dry ice. It would make this a bit easier logistically.

Line 90 states the method of dissection. There are a lot of historical papers describing these protocols which would be good to cite. Did you measure how much digesta was collected in mg? Did you measure the mass of the tissue which was processed through extraction? These could be important for your diversity estimates. If stored in etoh, how exactly did you then extract the sample… did you for instance add the entire volume of etoh or just some volume/biomass.

Lastly, the authors do not describe what part of the gut they actually sampled. Because the length of the fish aren’t described its hard to know the extent this matters but its important to describe if the whole gut vs partial was used. If partial was it towards stomach or anus. The site of the GI will differ in the microbiome.

Line 97: what were the modifications?
Line 98: extraction blanks or negatives aren’t very useful as they will primarily consist of ASVs from neighboring samples rather than background contaminants in a kit. (PMID: 31239396)

Line 146: The decontam pipeline, although published, severely overestimates the contaminants. It will remove many ASVs which are actually important to the outcomes. To ensure the results are robust, authors should reprocess and rerun stats on the dataset without removing these putative contaminants. Please provide details on what these actual contaminants were and their prevalence in the samples. Assuming this was supplemental table 9, you can see clearly that most of these organisms are common aquatic or fish digesta associated and likely should not be removed.

Line 169: how did you account for phylogenetic relatedness?

Line 172: how were the outliers determined?

Line 184: To compare compositional differences, authors only report using jaccard. I would recommend including other distances such as bray-curtis. I would also recommend and am unsure why they opted on not including phylogenetic measures such as unweighted and weighted unifrac.

Line 224: Its unclear if the authors rarified these samples. If they did not, they should justify this in the methods. Sequencing depth is a function of starting biomass which may not have been controlled.

Validity of the findings

The goal of this paper “Gut microbiota differences between paired mucus and digesta samples in three small species of fish” was to investigate and evaluate a new method for comparing the transient vs resident microbiome of the fish gut from small fish. The authors used 16S and compared microbiomes between 3 species of fish where were sampled from a total of 11 locations in Spain. Each fish was processed for 2 sample types: digesta and mucus.

Overall the findings are inline with the methods. I do however believe that some of the findings could change depending on the distance measure used along with the re-inclusion of ASVs which were removed by decontam.

Any reason why V hispanica was less differentiated by sample type and location?
Notes to editor

---

## Round 0.2 · accepted · Accept

Thank you for addressing all the comments.

·

Basic reporting

no comment

Experimental design

no comment

Validity of the findings

no comment

Additional comments

no comment